# Medication Adherence and Its Influencing Factors among Patients with Heart Failure: A Cross Sectional Study

**DOI:** 10.3390/medicina59050960

**Published:** 2023-05-16

**Authors:** Mohamad Jarrah, Yousef Khader, Osama Alkouri, Ahmad Al-Bashaireh, Fadwa Alhalaiqa, Ameena Al Marzouqi, Omar Awadh Qaladi, Abdulhafith Alharbi, Yousef Mohammed Alshahrani, Aidah Sanad Alqarni, Arwa Oweis

**Affiliations:** 1Department of Internal Medicine, Jordan University of Science and Technology, P.O. Box 3030, Irbid 22110, Jordan; mijarrah@just.edu.jo; 2Department of Public Health, Community Medicine, Jordan University of Science and Technology, P.O. Box 3030, Irbid 22110, Jordan; 3Faculty of Nursing, Yarmouk Univerity, P.O. Box 566, Irbid 21163, Jordan; 4Higher Colleges of Technology, Abu Dhabi P.O. Box 25026, United Arab Emirates; 5Philadelphia University, Amman 19392, Jordan; 6College of Health Sciences, University of Sharjah, Sharjah P.O. Box 27272, United Arab Emirates; 7College of Nursing, King Saud University, P.O. Box 11451, Riyadh 11451, Saudi Arabia; 8College of Nursing, University of Hail, P.O. Box 2440, Hail 81451, Saudi Arabia; 9Clinical Technology Department, Umm Al-Qura University, Mecca 24382, Saudi Arabia; 10College of Nursing, King Khalid University, Abha 61421, Saudi Arabia; 11Faculty of Nursing, Jordan University of Science and Technology, P.O. Box 3030, Irbid 22110, Jordan

**Keywords:** adherence, influencing factors, heart failure, medication

## Abstract

*Background and objectives*: The chronic nature of heart failure requires long-lasting consumption of various medications. Despite the therapeutic benefits of heart failure medications, about 50% of patients with heart failure don’t adequately adhere to their medications as prescribed globally. This study aimed to determine medication adherence levels among Jordanians with heart failure and its influencing factors. *Materials and Methods*: A cross-sectional study was conducted among 164 patients with heart failure attending cardiac clinics in the north of Jordan. The Medication Adherence Scale was used to measure medication adherence. *Results*: Overall, 33.5% of patients had high adherence, and 47% had partial to poor adherence. The proportion of patients with good to high adherence was significantly higher among patients younger than 60 years, having >high school level of education, being married, living with somebody, and having insurance. *Conclusions*: Patient-centered approach, targeting age, level of education, marital status, and health insurance coverage, should be developed using evidence-based guidelines to enhance adherence to medication and health outcomes in Jordanian patients with heart failure. The development and implementation of new and feasible strategies, particularly suited to Jordan’s healthcare system capabilities, is important to improve medication adherence.

## 1. Introduction

Heart failure (HF) is a chronic cardiac disease resulting in increased morbidity, hospitalizations, and deaths [1,2]. Patients with HF experience debilitating congestion symptoms, such as dyspnea, fatigue, cough, and ankle swelling [3]. Worldwide, approximately 64.34 million people (8.52 per 1000 inhabitants) live with HF, making up 346.17 billion US $ healthcare expenditure [4]. Despite the significant advancements in therapeutic directions, HF prognosis is still markedly poor worldwide, especially in developing countries [5,6]. The chronicity of heart failure requires long-lasting consumption of various medications to reduce morbidity and mortality [6,7]. Medication is a pivotal component of HF treatment, and thus patient adherence to medication is a crucial self-care behavior [8] to achieve the utmost therapeutic benefits of medication [2,6]. According to the Centers for Disease Control and Prevention [9,10], adherence to medication reflects the degree to which patient behavior, including taking medications, conforms to health instructions from healthcare providers, while nonadherence refers to patients who don’t take their medication as prescribed, or follow the recommendations of healthcare providers regarding taking medication.

The literature showed numerous benefits relating to medication adherence. Previous studies found that adherence is essential to maintain the physiologic functions of patients [1] and improve quality of life (QOL) and reduce hospitalizations [1]. Furthermore, medication adherence assists with managing symptoms, preventing deterioration, and improving clinical outcomes [2,6]. The American College of Cardiology, the American Heart Association, and the European Society of Cardiology guidelines emphasize the importance of adherence to medications due to their remarkable effect in reducing morbidity and mortality in HF [7,11].

A systematic review and meta-analysis of controlled trials also showed that poor medication adherence in HF is prevalent and leads to increased hospital admissions, mortalities, and unessential healthcare expenses irrespective of the etiology [2]. For example, a recent study indicated that non-adherent patients represented 22.1% of all hospital admissions associated with HF, and had more frequent hospital readmission (hazard ratio [HR] 0.45; 95% confidence interval [CI]: [0.25; 0.52]) [12]. However, adherence to HF medications is significantly low, adversely influencing clinical outcomes and resulting in worse heart failure, poor physical function, and greater risk for hospitalizations and mortality [8]. Additionally, sufficient adherence to HF medication is challenging for patients, and burdensome on their families as well as healthcare providers. Thus, adherence to medication remains below/under the levels required and desired [7]. Previous studies have shown that nearly 25% of patients with HF were not consistently adherent to their prescribed medications [13,14]. For frequency, roughly 50% of the patients affected with chronic HF take insufficient dosages of medication [1,6].

Various factors have been found to influence medication adherence in HF and other chronic diseases including gender, level of education, marital status, age, monthly income, medications cost, adverse effects of medications, family support, pill burden, and multiple comorbidities [3,15,16]. Additionally, race, body mass index (BMI), region, smoking, symptoms absence, cognitive impairment, lack of support, depression, poor attention and knowledge about medication, multiple medications, and difficulty swallowing were significant predictors of medication adherence in HF [17,18]. For example, being obese or overweight was associated with poor medication adherence [17,18]. Therefore, patients’ demographic and clinical characteristics that may influence adherence can be utilized to customize interventions aiming at improving medication adherence and ultimately improving self-care, QOL, and clinical outcomes.

In Jordan, data specifically targeting medication adherence in HF is scarce. However, risk factors for HF including, ischemic heart disease, obesity, smoking, diabetes, hypertension, and poor physical activity are significantly prevalent in Jordan [19]. This study aimed to determine medication adherence levels among Jordanians with heart failure and its associated factors. The findings of this study may assist healthcare providers in developing and implementing new and feasible strategies, particularly suited to Jordan’s healthcare system capabilities, to improve medication adherence among heart failure patients.

## 2. Methods

### 2.1. Design and Sampling

A cross-sectional survey was conducted among patients with heart failure attending a cardiac clinic in a teaching hospital in the north of Jordan. A convenience sample of 164 Jordanian patients with chronic HF attending cardiac clinics was selected.

The sample size was calculated assuming that 50% of patients have poor adherence at a level of significance of 0.05 and a margin of error of 10%. The calculated minimum sample size was 96 patients. The inclusion criteria were patients aged 18 years or older diagnosed with HF by a cardiologist through the New York Heart Association (NYHA) criteria, signs and symptoms of HF, and echocardiograph; and able to read and write Arabic. The exclusion criterion was the inability to complete the survey due to conditions such as severe dementia. Illiterate participants were included if they have a consenting companion who can read the survey for them to assist in completing it.

### 2.2. Ethical Consideration

Ethical approval was obtained from the Institutional Review Board at the hospital where the study was performed. In the current study, we followed the Helsinki Declaration. The invitation letters with the contact details of the primary investigator (PI) were distributed to the nurses who worked at outpatient clinics. The expected participants called the PI who arranged an interview with them. After answering the expected participants’ questions the PI explained further about the study’s purpose and confirmed maintaining their rights (e.g., that their responses would be anonymous and kept confidential, and that they can choose not to participate without their care being affected). A written consent form was signed by all patients. This consent implies the voluntariness of participation in filling out the questionnaire and the right to access medical records to collect the clinical data.

### 2.3. Instruments

The questionnaire contained two parts; the demographic sheet to collect patients’ characteristics and the General Medication Adherence Scale (GMAS) to GMAS to measure medication adherence in patients with chronic diseases in several studies [20,21,22]. The GMAS scale consists of 11 multiple-choice items, distributed across three domains relating to non-adherence including patient behavior, pill burden, additional disease, and expenditures. The scoring criteria include diverse adherence levels: high (30–33), good (27–29), partial (17–26), low (11–16), and poor (≤10) [20,21,22]. The tool was recently translated into Arabic language and validated in Saudi patients with chronic diseases, the reliability was high (Cronbach’s α of 0.865) [20,23]. For our study, translation from English to Arabic and back translation to English was performed to assess the coherence between the two versions. Additionally, the tool was piloted on ten patients with heart failure (six males and four females) who attended outpatient cardiac clinics and met the same inclusion criteria. The tool was found to be readable, clear, understandable, and. culturally appropriate.

### 2.4. Data Collection Procedure

On the day of the cardiac care clinic, the PI contacted the clinic nurses before the clinic starts to debrief them about the study, including the purpose and the ethical approval of the study. The clinic nurse was asked to distribute the invitation letter to the expected participants. Those who agreed and signed the informed consent met the data collector in a waiting room at the outpatient clinic. The nurse was also asked to complete the clinical information on the questionnaire for each participant agreed to participate.

After completing the survey, the researcher asked the participants to place it in the confidential collection box provided inside the meeting room. Then, the researcher, in collaboration with cardiac clinic nurses, accessed the participants’ medical records and filled the clinical information section.

### 2.5. Variable Definition

According to the United Nations age above 60 years was defined as older age [24]. For monthly income, the cutoff value of 350 JOD was defined as the minimum wage in Jordan [25]. The ejection fraction was dichotomized using a cutoff value of 55% for statistical purposes. The ejection fraction of less than 55% was defined as heart failure.

### 2.6. Data Analysis

The data were analyzed using the Statistical Package of the Social Sciences Program (SPSS version 21). Descriptive statistics including percentages, means, frequency, and standard deviation were used to describe adherence levels and to describe the demographic and clinical characteristics of the sample. The proportions of patients with good to high adherence to medication were compared according to socio-demographic and clinical characteristics using the Chi-square test. The multivariate analysis of factors associated with adherence to medication was conducted using binary logistic regression. Only significant variables remained in the model. A *p*-value of less than 0.05 was considered statistically significant.

## 3. Results

A total of 164 patients were included and completed the questionnaire in this study. Their age ranged from 32 to 82 years with an average of 62.5 (SD 8.8). Almost two-thirds (61.6%) of patients were males, 88.4% were married, and 47.0% had an education level of more than a high school education. Table 1 shows the demographic and clinical characteristics of patients.

Overall, 33.5% of patients had high adherence, 19.5% had good adherence, 20.7% had partial adherence, 9.8% had low adherence, and 16.5% had poor adherence. Table 2 shows the overall adherence to medication, non-adherence due to patient behavior, comorbidity and pill burden-related non-adherence, and cost-related non-adherence. The proportion of patients with good to high adherence was significantly higher among patients younger than 60 years, having income >350 JD, having >high school level of education, being married, living with somebody, having insurance, having no difficulty in sleeping, and having never been admitted to hospital for heart failure. Table 3 shows the proportion of patients with good to high adherence to medication according to socio-demographic and clinical characteristics.

In the multivariate analysis (Table 4), the only variables that remained significantly associated with adherence were age, education level, marital status, insurance, and admission to the hospital from heart failure. The odds of good to high adherence were significantly higher for patients 60 years old or younger compared to patients older than 60 years (OR = 3.1). Having an education higher than high school was associated with increased odds of good to high adherence (OR = 2.6). Married patients were more likely to highly adhere to medications compared to single patients (OR = 14). Insured patients were more likely than patients with no insurance to have good to high adherence (OR = 5.8). No admission to the hospital for heart failure in the past was associated with good to high adherence to medications (OR = 7).

## 4. Discussion

The present study determined the total medication adherence and its influencing factors among Jordanian patients with HF. The overall medication adherence was rather suboptimal in this study as the results indicated that approximately half of the patients had partial to poor adherence. It has been found that patients with chronic diseases, particularly in developing countries, were more likely to have lower medication adherence compared to others [26]. Our finding is confirmed by a systematic review which showed that the mean prevalence of non-adherence to hypertensive medication in developing countries was about 47.34% [27]. The review also identified various factors that could influence medication adherence such as monthly income, socioeconomic status, medication cost, and knowledge [27].

Previous studies from Jordan, targeting patients with chronic diseases such as cardiovascular diseases and diabetes, reported that the range of adherence level was between 31% to 46%, and found a variety of its associated factors including age, level of education, difficulty dispensing a prescription on time, higher number of prescriptions, and psychological distress (such as anxiety and depression) [28,29]. However, the varied levels of medication adherence reported in these studies can be attributed to differences in participants’ characteristics’, instruments used definitions of adherence, and methodologies. The other possible rationale for lower adherence level in Jordanian patients with HF might be related to a significant lacking in the structure of general prescriber/family doctor, in which about 40% of patients prescribe their medications, and up to 50% buy their medications from pharmacies without receiving counseling of physician or pharmacist [29].

Our study found that the odds of good to high adherence were significantly higher for patients aged 60 years or younger. The literature showed that older age is associated with lower medication adherence due to several factors including the burden of associated comorbidities, multiple prescriptions, forgetfulness, and financial hardship [6,26,30,31]. Interventions, tailored to patients with HF aged 60 years or older such as education before hospital discharge, reminders through follow-up phone calls, and in-person teaching sessions that can be conducted in community healthcare centers, can assist with enhancing medication adherence and health outcomes [31].

The present study showed that married patients had higher medication adherence compared to those unmarried. Patients living with HF mostly need assistance from their spouses, to adhere to prescribed medication such as money to afford the prescription, transportation to a cardiac clinic to update/adjust medication, direct supervision, reminders to take the medication in a timely manner, and support to surmount fatigue and any mental changes that may affect adherence to their medications as prescribed). For example, a study reported that married patients frequently reported needs from their spouses to remind or assist them with taking their medications. Additionally, married patients with HF expressed that they received more social support compared to unmarried, which resulted in greater medication adherence [32,33,34,35,36,37]. Therefore, assessing marital status in patients with HF can enable healthcare providers to determine those at higher risk for poor medication adherence and worse clinical outcomes. Furthermore, tailoring interventions for unmarried patients can enhance medication adherence and clinical outcomes.

The present study indicated that patients with higher educational levels were more adherent to their prescribed medication. The reason for this finding can be explained that patients with higher education have a higher ability to communicate with healthcare providers and read/comprehend educational material, which eventually helps in acquiring greater health knowledge and enhancing medication adherence. This result is in line with findings from previous studies [26,38,39]. Therefore, the patient’s level of education should be determined and documented, to help healthcare providers in tailoring the most efficacious interventions and special educational material based on their learning needs.

We found that having health insurance is associated with higher medication adherence. This finding is congruent with other results of other studies targeting chronic diseases such as diabetes and hypertension [6,40]. Jordan still has inadequate health insurance coverage (nearly 69% of the citizens, 68% for males, 70% for females), due to various problems and challenges such as increased medication costs, insufficient funds, and large reflux of refugees to Jordan [41]. A study supports our finding which indicates that in Jordan, as a low-income country, medication cost was considered a major problem for patients living with chronic diseases especially those elderly, uninsured, have a low level of education, and are prescribed multiple medications [42].

The establishment of educational programs, rehabilitation programs, and tracking systems can raise awareness of the importance of adherence to therapeutic regimens [43]. Our findings also highlight the need to design simple and clear reminders and audio-visual materials, particularly for patients with lower education and those aged over 60, to assist with improving their health knowledge and medication adherence. The Jordan Ministry of Health in collaboration with other health organizations should develop new health policies to achieve universal health coverage and to increase the affordability of medication costs among uninsured people. The results of the current study could help in updating educational interventions to improve medication adherence among heart failure patients and their caregivers. Furthermore, conducting longitudinal and mixed studies may provide a holistic image and deep insight into factors that affect patients’ adherence to medications. Emphasizing the importance of maintaining adherence to medication in nursing curricula could help patients avoid complications and death.

This study has some limitations. Using self-reported questionnaires to collect data from a single geographical area using a cross-sectional design might limit the generalizability of the study. Additionally, we did not address the effect of adverse events associated with drug administration on adherence levels. Some information needed for a comprehensive understanding of factors that influence medication adherence is insufficient such as New York Heart Association (NYHA) Functional Classification, number of drugs per day, other medications to take for co-existing diseases, and how long the included patients were diagnosed with heart failure. Some patients in Jordan have health insurance that does not pay 100% of the bill; patients are obliged to pay part of the treatment expenses. This issue has not been examined in this study. This study examined only outpatient criteria and could not investigate inpatients’ experience nor asked about the type of health insurance they had (whether full or partial). Inpatient criteria are also important aspects that influence adherence among HF patients that could be studied in future studies. Therefore, prospective and experimental studies are warranted to emphasize and measure the effect of selected interventions that would improve adherence among HF patients. More factors, which may influence medication adherence, should be investigated in future studies such as the number of pills consumed per day and other medications to take for co-existing diseases. Furthermore, future research studies are recommended to measure biochemical data such as plasma CRP, IL-6, TNF, hs-CRP, VEGF, and nitric oxide levels to identify how they change and when they change in those who are taking medicines and not taking medicines regularly to explain their lack of response or worsening of the condition or benefit from drug therapy.

## 5. Conclusions

The findings of this study provide insight relating to medication adherence and its associated factors in Jordanian patients with HF. Poor adherence to HF medication exerts/imposes a significant burden on the healthcare system and eventually results in increased hospital readmissions and deaths. Measures for the assessment and improvement of medication adherence should be considered an integral component of HF care. Patient-centered approach, targeting age, level of education, marital status, and health insurance coverage, should be developed using evidence-based guidelines to enhance adherence to medication and health outcomes in patients with HF. Moreover, strategies to expand health insurance coverage are crucial in improving adherence, particularly among people living in a low-income country such as Jordan. The results of the study can be useful for policymakers to improve patient health outcomes and decrease the cost of care in the future by addressing these factors in healthcare planning. Additionally, our results could inform designing intervention studies to develop multidimensional interventions aimed at enhancing and promoting medication adherence in patients with HF. Such interventions may focus on (1) promoting social support for the use of medication within participants’ social networks and (2) building a supportive relationship between healthcare providers and patients to improve health outcomes through health education and promotion.

## Figures and Tables

**Table 1 medicina-59-00960-t001:** Participants’ demographic and clinical characteristics (n = 164).

Variable	n (%)
Age	
≤60	77 (47)
>60	87 (53)
Gender	
Male	101 (61.6)
Female	63 (38.4)
Marital status	
Single	19 (11.6)
Married	145 (88.4)
Education	
≤High School	87 (53)
>High School	77 (47)
Monthly income (Jordanian Dinars)	
≤350	75 (45.7)
>350	89 (54.3)
Primary caregiver	
Patient	106 (64.6)
Family member	58 (35.4)
Living alone	24 (14.6)
Have insurance	132 (80.5)
Smoking	54 (32.9)
Have difficulty sleeping	83 (50.6)
Body Mass Index	
Normal	23 (14)
Overweight	70 (42.7)
Obesity	71(43.3)
Have ever been admitted to the hospital for heart failure	54 (32.9)
Diabetes Mellitus	45 (27.4)
Hypertension	125 (76.2)
Hypercholesterolemia	104 (63.4)
Acute coronary syndrome	81 (49.4)
Stroke	1 (0.6)
Ejection fraction	
<55	157 (95.7)
≥55	7 (4.3)

**Table 2 medicina-59-00960-t002:** The overall adherence to medication, non-adherence due to patient behavior, comorbidity and pill burden-related non-adherence, and cost-related non-adherence.

	Total Adherence		Non-Adherence Due to Patient Behavior (PBNA)		Comorbidity and Pill Burden-Related Non-Adherence (ADPB)		Cost-Related Non-Adherence (CRNA)	
Category	N	%	N	%	N	%	N	%
High Adherence	55	33.5	52	31.7	82	50.0	69	42.1
Good Adherence	32	19.5	37	22.6	20	12.2	37	22.6
Partial Adherence	34	20.7	25	15.2	27	16.5	24	14.6
Low Adherence	16	9.8	20	12.2	13	7.9	11	6.7
Poor Adherence	27	16.5	30	18.3	22	13.4	23	14.0

**Table 3 medicina-59-00960-t003:** The proportion of patients with good to high adherence to medication according to socio-demographic and clinical characteristics.

	Total Adherence				Sig.
	Partial or Worse		Good to High		
	N	%	N	%	
Age					<0.001
≤60	18	23.4	59	76.6	
>60	59	67.8	28	32.2	
Gender					0.648
Male	46	45.5	55	54.5	
Female	31	49.2	32	50.8	
Marital Status					0.001
Single	16	84.2	3	15.8	
Married	61	42.1	84	57.9	
Educational Level					<0.001
≤high school	53	60.9	34	39.1	
>high school	24	31.2	53	68.8	
Income					<0.001
≤350	47	62.7	28	37.3	
>350	30	33.7	59	66.3	
Caregiver					0.465
Patient	52	49.1	54	50.9	
Family member	25	43.1	33	56.9	
Living alone					<0.001
Yes	20	83.3	4	16.7	
No	57	40.7	83	59.3	
Having insurance					<0.001
Yes	50	37.9	82	62.1	
No	27	84.4	5	15.6	
BMI					0.333
Normal	10	43.5	13	56.5	
Overweight	29	41.4	41	58.6	
Obesity	38	53.5	33	46.5	
Ejection fraction					0.581
<55	73	46.5	84	53.5	
≥55	4	57.1	3	42.9	
Smoking					0.122
Yes	30	55.6	24	44.4	
No	47	42.7	63	57.3	
Having difficulty sleeping					<0.001
Yes	51	61.4	32	38.6	
No	26	32.1	55	67.9	
Have ever been admitted to the hospital for heart failure					<0.001
Yes	42	77.8	12	22.2	
No	34	31.2	75	68.8	
Diabetes Mellitus					0.006
Yes	29	64.4	16	35.6	
No	48	40.3	71	59.7	
Hypertension					0.630
Yes	60	48.0	65	52.0	
No	17	43.6	22	56.4	
Hypercholesterolemia					0.175
Yes	53	51.0	51	49.0	
No	24	40.0	36	60.0	
MI or Angina					0.214
Yes	42	51.9	39	48.1	
No	35	42.2	48	57.8	
Stroke					0.345
Yes	0	0.0	1	100.0	
No	77	47.2	86	52.8	

**Table 4 medicina-59-00960-t004:** The multivariate analysis of factors associated with adherence to medication among patients.

Variable	OR	95% Confidence Interval		*p*-Value
Age (≤60 vs. >60)	3.1	1.3	7.5	0.009
Educational Level (>High School vs. ≤High School)	2.6	1.1	6.2	0.037
Marital Status (Married vs. Single)	14.0	3.1	62.8	0.001
Having insurance (Yes vs. No)	5.8	1.8	18.5	0.003
Ever admitted to hospital for heart failure (Yes vs. No)	7.0	2.8	17.6	<0.001

## Data Availability

The data that support the findings of this study are available upon reasonable request.

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
