# Peer review of "Medication Adherence and Its Influencing Factors among Patients with Heart Failure: A Cross Sectional Study"

_medicina, 2023, doi:10.3390/medicina59050960_

Round 1
Reviewer 1 Report
Dear authors,
Thank you very much for the opportunity to review the manuscript Medication adherence and its influencing factors among patients with heart failure: A cross-sectional study. The topic is very actual. The available data show that poor medication adherence is associated with worse health outcomes, including higher hospitalizations and death rates in patients with HF.
Below are my comments to improve your manuscript:
Please give a proper definition and citation for adherence/nonadherence (i.e. WHO)
In the methodology section, please give information about the reliability of GMAS for your study.
Please explain the definition of all variables used in the analysis.
1. The age >60 <60
2. Monthly income ≤ 350 >350
3. BMI classification
4. Ejection fraction <55 >55. However, please note that according the guidelines, the HF according to the EF is divided into: HFeRF <=40; HRmEF 41-49%, HFpEF >=50%.
5. There is lack of NYHA scale
Did you ask patients about the number of drugs for polypharmacy? Also, there is a reason for poor adherence.
Please describe the statistical methods - did you check the normality of the data and choose the appropriate statistical test?
Reviewer 2 Report
This study investigated medication adherence levels among Jordanians with heart failure and its associated factors. The authors found that 33.5% of patients had high adherence, and 47% had partial to poor adherence. The proportion of patients with good to high adherence was significantly higher among patients younger than 60 years, having at least a high school education, being married, living with somebody, and having insurance. I have some comments for the authors.
1. Was the information about how many pills per day the included patients had to take also collected? Did included patients also have other medications to take for co-existing diseases? If so, please add the information and perform an analysis or some discussion.
2. Is there any sex difference regarding medication adherence among the included patients?
3. Was the information about how long the included patients were diagnosed with heart failure also collected?
4. The significant limitations of the study are 1) it was done at a single healthcare institution and 2) the information needed for a comprehensive understanding of factors that influence medication adherence is insufficient. The authors may consider discussing more the limitations and further research plans.
